# Photosynthetic and Morphological Responses of Sacha Inchi (*Plukenetia volubilis* L.) to Waterlogging Stress

**DOI:** 10.3390/plants11030249

**Published:** 2022-01-18

**Authors:** Chyi-Chuann Chen, Ming-Sheng Li, Kuan-Ting Chen, Yueh-Hua Lin, Swee-Suak Ko

**Affiliations:** 1Biotechnology Center in Southern Taiwan, Academia Sinica, Tainan 711, Taiwan; luffa@gate.sinica.edu.tw (C.-C.C.); lms@gate.sinica.edu.tw (M.-S.L.); lyhlyh510216@gmail.com (Y.-H.L.); 2Department of Horticulture and Landscape Architecture, National Taiwan University, Taipei 10617, Taiwan; tim610565@gmail.com; 3Agricultural Biotechnology Research Center, Academia Sinica, Taipei 115, Taiwan

**Keywords:** CO_2_ uptake, climate change, waterlogging stress, hypoxia, chlorophyll florescence, sustainable agriculture, sacha inchi

## Abstract

Sacha inchi (*Plukenetia volubilis* L.) is an important oilseed crop that is rich in fatty acids and protein. Climate-change-related stresses, such as chilling, high temperature, and waterlogging can cause severe production loss in this crop. In this study, we investigated the photosynthetic responses of sacha inchi seedlings to short-term waterlogging and their morphological changes after long-term waterlogging stress. Sacha inchi CO_2_ uptake, stomatal conductance, and transpiration rate are affected by temperature and light intensity. The seedlings had a high CO_2_ uptake (>10 μmol m^−2^s^−1^) during the daytime (08:00 to 15:00), and at 32 and 36 °C. At 32 °C, CO_2_ uptake peaked at irradiations of 1000 and 1500 µmol m^−2^s^−1^, and plants could still perform photosynthesis at high-intensity radiation of 2000–3000 µmol m^−2^s^−1^. However, after 5 days of waterlogging (5 DAF) sacha inchi seedlings significantly reduced their photosynthetic ability. The CO_2_ uptake, stomatal conductance, Fv/Fm, ETR, and qP, etc., of the susceptible genotypes, were significantly decreased and their wilting percentage was higher than 50% at 5 DAF. This led to a higher wilting percentage at 7 days post-recovery. Among the four lines assessed, Line 27 had a high photosynthetic capability and showed the best waterlogging tolerance. We screened many seedlings for long-term waterlogging tolerance and discovered that some seedlings can produce adventitious roots (AR) and survive after two weeks of waterlogging. Hence, AR could be a critical morphological adaptation to waterlogging in this crop. In summary, these results suggest that improvement in waterlogging tolerance has considerable potential for increasing the sustainable production of sacha inchi.

## 1. Introduction

Sacha inchi (*Plukenetia volubilis* L.), or Inca peanut, originates from the Amazonian tropical rain forest. It has been consumed for thousands of years. Sacha inchi is a woody vine that produces large seeds containing high levels of polyunsaturated fatty acids, such as omega-3 and omega-6 lipids, as well as high levels of proteins, phenolic compounds, and antioxidants, which are beneficial for health [1,2]. It has been claimed that the economic value of sacha inchi is enormous, but it is thus far underutilized and deserves more research attention [3].

Environmental factors, such as light, temperature, water, etc., affect the photosynthesis rate of plants [4]. Abiotic stresses reduce photosynthesis capacity due to the inhibition of chlorophyll biosynthesis, the performance of photosystems, electron transport mechanisms, stomatal conductance, and transpiration rate, among many other negative consequences [5]. Sacha inchi can grow across a wide range of temperatures, i.e., between 10 and 37 °C, but temperatures below 8 °C cause cold stress. In a prior study, the maximum quantum efficiency of photosystem II (Fv/Fm) decreased after chilling treatment. Sacha inchi was found to be more susceptible to chilling stress than two other oilseed crops, *Jatropha curcas* and *Ricinus communis* after 3 days of chilling treatment [6]. Sacha inchi prefers a high light growing environment. Low light or shaded conditions delay flowering and decrease seed yield [7]. Plants grown under 100% light showed a higher photosynthesis rate and higher biomass compared to those grown under low-light treatment (20% or 52% light).

Climate change is associated with extreme weather events, such as long periods of heavy rain leading to soil flooding, which negatively affects plant growth [8]. Oxygen and CO_2_ gas exchange via the stomata and cell walls are limited underwater in flooding conditions. The low CO_2_ in flooded leaves restricts photosynthesis, and long-term flooding causes an energy crisis in plants [9]. In addition, hypoxia causes a lack of oxygen, inhibiting root respiration and growth, thereby restricting plant growth [10]. In well-aerated field conditions, soil redox (E_h_) ranges from 400~700 mV, which favors oxidation and aerobic microorganism growth. However, waterlogging decreases E_h_, reduces the soil oxygen concentration and represses soil microhabitats [11].

It was found that a flood-sensitive soybean cultivar had significantly lower chlorophyll content, photosynthesis, transpiration, and Fv/Fm than a tolerant cultivar under flooding conditions [10]. Therefore, maintaining high levels of photosynthesis is important for plant survival under flooding stress [12]. Furthermore, plant morphological changes, such as the formation of adventitious roots (ARs), aerenchyma, and radial O_2_ loss barriers are survival strategies to adapt to flooding stress [13]. The expansion of adventitious roots helps to improve the oxygenation of submerged tissues [14]. The aerenchyma contains air cavities, which increase oxygen transportation and the formation of a barrier against radial oxygen loss, thus enhancing plant tolerance to flooding [15,16]. Perennial pepperweed (*Lepidium latifolium*) is a unique species that is adapted to flooding, as well as drought conditions; therefore, its anatomy has attracted research attention. Pepperweed tends to reduce its root/shoot ratio, produce adventitious rooting, and develop aerenchyma under flooding conditions [17]. These traits are vital for the plant to adapt to flooding stress. Continuous hypoxic stress increases reactive oxygen species (ROS) accumulation and formation of autophagy-related structures in wheat roots, where cell autophagy first occurs in the stele and then in the cortex [18]. Therefore, the role of antioxidants and ROS scavengers is crucial for plant survival under flooding conditions [19].

Sacha inchi is an oil crop with considerable potential and high economic value [3]. Taiwan just started planting this crop in 2015, and the planting area now exceeds 1200 hectares. Field investigations have found that high temperature and continuous heavy rains or typhoons in summer caused floods and often caused the sacha inchi plants in the orchard to die due to waterlogging. In this study, we characterized the photosynthetic behavior of sacha inchi under different conditions (temperature, light), and analyzed the relationship between photosynthetic capacity and waterlogging tolerance. We also screened seedlings under long-term waterlogging conditions and investigated the morphological changes.

## 2. Results

### 2.1. Variation of Photosynthesis Rate

To understand the potential of CO_2_ uptake and stomatal conductance of sacha inchi, we used a Li-Cor 6800 photosynthesis system to detect CO_2_ uptake and stomatal conductance of two sacha inchi lines (Line 08-10 and Line 07) under constant 1000 µmol m^−2^s^−1^ irradiation to avoid sunlight fluctuation in the greenhouse. The results indicated that Line 07 has higher CO_2_ uptake and stomatal conductance (gs) than Line 08-10, but both lines have a similar changing trend (Figure 1). CO_2_ uptake of Line 07 was moderately high at 06:00 (8 μmol m^−2^s^−1^), high at 08:00 to 15:00 (12.5 μmol m^−2^s^−1^), and then declined to less than 3 μmol m^−2^s^−1^ at 20:00 (Figure 1a). The stomatal conductance of sacha inchi was less than 0.1 mol m^−2^s^−1^ in the early morning and after 18:00. During the daytime from 09:00 to 14:30, it had a high stomatal conductance value (>0.3 mol m^−2^s^−1^) (Figure 1b). These results indicate that the stomatal opening of sacha inchi is affected by circadian rhythm, which, therefore, affects the CO_2_ uptake. Moreover, there is genetic variation in photosynthesis capacity. 

### 2.2. Effect of Temperature on Photosynthesis

We quantified the photosynthetic rate of sacha inchi seedlings at various temperatures but fixed the light intensity in the Li-Cor chamber to a constant value of 1000 µmol m^−2^s^−1^, with 60% RH, and 400 µmol mol^−1^ CO_2_. Sacha inchi has its highest CO_2_ uptake at 36 °C; moderately high uptake at 28 °C, 32 °C, and 40 °C; but decreased uptake at 25 °C and 42 °C (Figure 2a). The seedlings showed maximum stomatal conductance at 32 °C and 36 °C, and the lowest at 25 °C (Figure 2b). The transpiration rate of sacha inchi was low at 25 °C but increased with increasing temperature to a maximum of 42 °C (Figure 2c).

### 2.3. Effect of Light Intensity on Photosynthesis

Our previous experiment showed that sacha inchi seedlings had a high CO_2_ uptake and low transpiration rate at 32 °C (Figure 2a,c); therefore, we fixed the chamber conditions to 32 °C, 60% RH, and 400 µmol mol^−1^ CO_2_ to test the effect of various light intensities on the photosynthesis rate of sacha inchi seedlings. Under dark conditions, sacha inchi did not uptake CO_2_ at all. Once irradiated, it exhibited quick uptake of CO_2_. It had a maximum CO_2_ uptake at 1000 and 1500 µmol m^−2^s^−1^ irradiation. Surprisingly, sacha inchi sustained moderately high CO_2_ uptake at a strong light intensity of 3000 µmol m^−2^s^−1^ (Figure 3a). Stomatal conductance increased at 1000 and 1500 µmol m^−2^s^−1^, but decreased when irradiation was ≥2000 µmol m^−2^s^−1^ (Figure 3b). The changing trend in the transpiration rate is similar to that of stomatal conductance. Sacha inchi showed the lowest transpiration rate under dark conditions, followed by irradiation at 500 µmol m^−2^s^−1^. However, it had a high transpiration rate at 1000 to 3000 µmol m^−2^s^−1^ irradiation (Figure 3c). These results indicate that sacha inchi has a high level of photosynthesis under strong light irradiation. It was surprising to find that sacha inchi can tolerate high-intensity radiation of 3000 µmol m^−2^s^−1^ and perform photosynthesis.

### 2.4. Photosynthetic Response of Different Genotypes to Waterlogging Stress

To determine photosynthesis of different sacha inchi genotypes to waterlogging stress, we tested four genotypes of seedlings (Lines 01, 07, 21, and 27). As shown in Figure 4, sacha inchi Line 07 had the highest SPAD, CO_2_ uptake, stomatal conductance, Fv’/Fm’ and ETR, but the lowest NPQ and qN compared to the other three genotypes before waterlogging. After 5 days of waterlogging, Lines 07 and 27 maintained higher SPAD, CO_2_ uptake, stomatal conductance, Fv/Fm, ETR, and qP than Lines 01 and 21 (Figure 4a–d,f–h). However, Lines 01 and 21 showed a complete loss of CO_2_ uptake and stomatal conductance at 5 DAF (Figure 4b,c). Among all the assessed genotypes, Line 01 exhibited severe wilting, and Line 21 had yellow leaves and some leaves fell at 5 DAF (Figure 4j, arrowheads). Conversely, Lines 07 and 27 showed less leaf wilting (Figure 4j). The leaf wilting percentage of Lines 01 and 21 was more than 50% but there was less wilting in Lines 07 and 27 at 5 DAF (Figure 4k). After recovery in the greenhouses for 7 days, only Line 27 had less than 50% leaf wilting. Statistical analysis showed that the wilting percentage of the four Lines was not significantly different between 5 DAF and after recovery for 7 days, however, the mean wilting percentage of the Lines 01, 07, and 21 were increased after recovery (Figure 4k).

### 2.5. Correlation between Waterlogging Traits and Photosynthesis Parameters

We conducted a Pearson correlation analysis to determine the relationship between the SPAD value, leaf wilting percentage (%), and the photosynthetic parameters at 5 DAF. As shown in Table 1, the leaf wilting percentage at 5 DAF was negatively correlated with SPAD value, stomatal conductance, CO_2_ uptake, transpiration rate, Fv/Fm, Fv′/Fm′, ETR, NPQ, and qP. However, qN was not significantly correlated (Table 1). These data suggest that high gas exchange and PSII are associated with low wilting percentage and less susceptibility to waterlogging damage.

### 2.6. Adventitious Root Formation of Sacha Inchi after Long-Term Waterlogging

To screen waterlogging-tolerant sacha inchi genotypes, we waterlogged seedlings for two weeks and investigated the morphological changes. We found that the primary and lateral roots and the submerged hypocotyls of the waterlogging-susceptible plants were rotten and the leaves were withered (Figure 5a). However, tolerant seedlings maintained green leaves and survived after long-term waterlogging. Although their primary and lateral roots were rotten, they developed new and healthy adventitious roots (white arrowhead) in the submerged hypocotyl zone near the water surface (Figure 5b). It seems that under long-term waterlogging stress conditions, the survival of sacha inchi pretty much relies on the development of a new root system (AR) to replace the defective original roots.

## 3. Discussion

### 3.1. Photosynthetic Characteristics of Sacha Inchi

To understand the photosynthetic behaviors of sacha inchi seedlings, we examined their gas exchange under different conditions, including temperature, light, and waterlogging stress. Our results indicated that sacha inchi seedlings have high stomatal conductance at moderately high temperatures from 32 to 36 °C (Figure 2b) and under 1000 to 1500 µmol m^−2^s^−1^ irradiation regimes (Figure 3b). Figure 3c shows that there was no significant change in the transpiration rate at 32 °C with exposure to different light intensities. The transpiration rate remained less than 4 mmol m^−2^s^−1^, even at high intensity irradiation (3000 µmol m^−2^s^−1^). However, temperature significantly affected the transpiration rate. When the temperature was higher than 36 °C, the transpiration rate was over 4 mmol m^−2^s^−1^ (Figure 2c). Presumably, the increased sacha inchi transpiration rate seen with high-temperature is mainly due to the effect of temperature-dependent cuticle transpiration as mentioned previously [20,21]. It was reported that when temperatures are higher than 35 °C, there is significantly increased cuticular water permeability. Therefore, to improve sacha inchi nursery quality, it is suggested that greenhouse environments are maintained at the above suitable conditions to increase seedlings’ photosynthesis rate and enhance growth. High transpiration might cause severe water loss. We found that the effect of temperature on transpiration of sacha inchi is more pronounced than light intensity. This may explain why, during summer, when there is waterlogging caused by heavy rain, there is tremendous wilting and death of sacha inchi in the field. The dual effects of root damage and leaf loss of water may lead to vulnerability.

### 3.2. Response of Sacha Inchi to Waterlogging Stress

Sacha inchi exhibited reduced photosynthesis at 5 DAF. SPAD and photosynthesis parameters, such as CO_2_ uptake, stomatal conductance, Fv′/Fm′, and ETR were significantly reduced, as shown in Figure 4. These results are consistent with a previous study of the effect of root flooding on field bean photosynthesis [22]. Pearson correlation analysis indicated that gas exchange (CO_2_ uptake, stomatal conductance, and transpiration rate) and PSII parameters (ETR, qP, NPQ, and Fv/Fm) were negatively correlated with the wilting percentage at 5 DAF (Table 1)**.** Waterlogging increases leaf wilting percentage and low gas exchange mainly due to the stomata being almost completely closed in the wilted leaves [23]. Subsequently, stomatal closure inhibits PSII function due to the decreased intercellular CO_2_ (Ci) and chlorophyll fluorescence parameters [24]. Liu et al. [25] studied the response of *Distylium chinense*, a flood-tolerant plant for vegetation recovery of the flood plain, to flooding and indicated that Fv/Fm, qP, and ETR are significantly decreased during the flooding period; however, qN was not significantly different for flooded and non-flooded plants. We found similar results in our study. However, Line 27 did not show significantly decreased Fv/Fm (Figure 4d), and Line 01 significantly increased qN after waterlogging (Figure 4i). Interestingly, Line 27 is a waterlogging-tolerant genotype, but Line 01 is a susceptible genotype in our study.

Based on the leaf wilting percentage after waterlogging (Figure 4j,k), we recommend that 50% of wilting at 5 DAF could be used as a “cut-off point” to differentiate tolerant and susceptible lines of sacha inchi seedlings at an early stage. Among the four genotypes tested, Lines 01 and 21 are waterlogging susceptible; whilst Line 27 was the most tolerant line with <50% wilting at 5 DAF and 7 days post-recovery in the greenhouse. After recovery, the leaf wilting percentage of Line 27 tends to decrease, and the seedlings can grow and produce new leaves. Similar to Line 27, Line 07 had a lower percentage of wilting at 5 DAF, but it wilted more than 50% post-recovery; hence, it could be grouped as having moderate tolerance. Overall, the order of waterlogging tolerance of these four genotypes is: Line 27 > Line 07 > Line 01 = Line 21. Except for Line 27, Lines 01, 07, and 21 showed severe wilting, which may be due to ROS injury, and their root function cannot be restored after recovery. Flooding recovery causes a sudden oxygen burst that accelerates oxidative stress, increases plant damage, and leads to rotten roots and wilting leaves had been reported [19,26].

After long-term waterlogging, we observed some tolerant seedlings produced aquatic adventitious roots at the submerged hypocotyl (Figure 5b). It appears that sacha inchi is similar to other crops, such as cotton, barley, and cucumber that develop new adventitious roots to alleviate waterlogging stress. Adventitious roots are essential to replace the original damaged primary root and lateral roots and help seedlings uptake water and nutrients under waterlogging conditions [23,24,27]. As mentioned before, well-aerated adventitious roots are good for promoting stomatal opening during long-term waterlogging [28].

### 3.3. Crop Improvement for Waterlogging Tolerance Sacha Inchi

Waterlogging stress causes severe production loss of sacha inchi. To solve this problem to achieve sustainable production, we propose breeding a waterlogging-tolerant variety. We suggest that the ideal phenotypes of a tolerant variety should include both physiological and morphological adaptations. From the physiological perspective, maintaining the balance between photosynthesis and water loss is essential for plant survival under the combination of high temperature and flooding stress. Moreover, adventitious root development is one of the morphological adaptation traits that enable sacha inchi to survive under long-term waterlogging stress. Sacha inchi seedlings with a high wilting percentage (susceptible line) were eliminated easily within one week after waterlogging. Those lines showing less wilting should be further waterlogged over a longer period to confirm the development of adventitious roots for isolating the tolerant lines. Considering the complex conditions in the field, multiple abiotic and biotic factors might accelerate the damage to sacha inchi under waterlogging. Therefore, in addition to abiotic stresses (high temperature and high light), more attention should be paid to the plant-microbe interactions.

## 4. Materials and Methods

### 4.1. Plant Materials and Experimental Design

The sacha inchi seeds used in this study were collected separately from the individual sacha inchi plants from a farmer’s field in Liujia District, Tainan, Taiwan (23°14′56.0″ N 120°19′32.5″ E). They were labeled Lines 08–10, 01, 07, 21, and 27. Seeds were surface-sterilized with 35 ppm hypochlorous acid water for 10 min, soaked in reverse osmosis (RO) water for 3 h, then placed in a plastic box containing wet filter paper, and seeds were germinated in the dark in an incubator set at 32 °C. Germinated seeds were transferred to a pot (7.5 cm width × 7.5 cm height) containing 140 mL peat moss. Seedlings were raised in the greenhouse of the Academia Sinica Biotechnology Center in Southern Taiwan (AS-BCST), Tainan, Taiwan with day and night temperatures of 32 ± 4 and 24 ± 4 °C, respectively. Sacha inchi seedlings at the five-leaf stage (approximately 40 cm height) were used for measuring photosynthesis and flooding experiments. We used high-yielding sacha inchi Lines 08–10 and Line 07 (good yielding and tolerant to waterlogging in the orchard) to study the variation of photosynthesis. Line 07 was used to study the response of photosynthesis to various temperatures and light intensities. Each treatment was tested on four seedlings. To evaluate the response of the four different genotypes of sacha inchi, i.e., Lines 01, 07, 21, and 27, to waterlogging, six seedlings per line were placed in a plastic box (24 cm height × 60 cm length × 43 cm width) and waterlogged at 5 cm above the soil surface. We observed some wilting symptoms at 5 days after waterlogging and evaluated the plants’ tolerance level. To measure chlorophyll content, we used a nondestructive soil plant analysis development (SPAD) chlorophyll meter (model SPAD-502) as described previously [29]. We recorded the wilting percentage of sacha inchi at 5 days after flooding and after recovery for 7 days in the greenhouse. The wilting percentage was recorded based on the percentage of wilting leaves per seedling. We performed long-term waterlogging for two weeks to screen promising tolerant genotypes. To screen the long-term waterlogging tolerance line, about 200 bulk seeds collected from the farmer’s field were sown, and the seedlings were waterlogged for two weeks. Morphological change and adventitious root development were recorded.

### 4.2. Measurement of Photosynthetic Parameters

We used a portable photosynthesis system, LI-6800F Gas Exchange and Fluorescence System (Li-Cor, Lincoln, NE, USA), to determine the gas exchange of sacha inchi including CO_2_ uptake (A), stomatal conductance (gs), and transpiration rate (E). To understand the changes of photosynthesis in sacha inchi, we recorded gas exchange data from 06:00 (PAR in the greenhouse was <10 µmol m^−2^s^−1^) to 20:00 (PAR in the greenhouse was <0.2 µmol m^−2^s^−1^). The Li-Cor chamber was set at 32 °C; 1000 µmol m^−2^s^−1^; 400 µmol mol^−1^ CO_2_; 60% RH. An autologging program was used to record data every 60 s. Based on the variation of photosynthesis, we know sacha inchi tends to close stomata after 17:00. Therefore, we recorded gas exchange measurements from 9:00 to 15:00 (average PAR in the greenhouse was 250 + 124 µmol m^−2^s^−1^). The measurements were taken on a newly full expanded leaf. To study the effect of temperature on photosynthesis, we fixed the Li-Cor chamber conditions at 1000 µmol m^−2^s^−1^, 60% RH, and 400 µmol mol^−1^ CO_2_, and set various temperatures of 25, 28, 32, 36, 40, and 42 °C. To study the effect of light intensity on gas exchange, we set the chamber temperature at 32 °C and used seven levels of irradiation, i.e., 0, 500, 1000, 1500, 2000, 2500, and 3000 μmol photons m^−2^s^−1^. We exposed newly established leaves to various temperature and light treatments for approximately 10 min until the CO_2_ uptake curve was stabilized, then data were recorded.

### 4.3. Measurement of Chlorophyll Fluorescence

We measured chlorophyll fluorescence with the portable photosynthesis system (LI-6800F, Li-Cor, Lincoln, NE, USA) as described previously [30]. The instrument was equipped with a fluorescence leaf chamber (6800-01A, Li-Cor), using a 6 cm^2^ aperture for the fluorometer. The CO_2_ concentration was set to a constant 400 µmol mol^−1^. The actinic light was composed of 90% red (625 nm) and 10% blue (470 nm). Light intensity was set to 1000 µmol mol^−1^. New fully established leaves of sacha inchi were dark-adapted for 2 h before measuring the parameters of chlorophyll fluorescence, and a saturating actinic light pulse of 8000 µmol m^−2^ s^−1^ was applied for 1 s to measure the maximal fluorescence. We collected data on the maximum fluorescence yield in dark-adapted leaves (Fv/Fm), electron transport rate (ETR), nonphotochemical quenching (NPQ), photochemical quenching of fluorescence (qP) and nonphotochemical quenching of fluorescence (qN). Each treatment was conducted on six seedlings.

### 4.4. Statistical Analyses

We performed statistical analyses using IBM SPSS Statistics software version 22. We applied the one-way analysis of variance (ANOVA) procedure, followed by Duncan’s multiple range test. The student’s *t*-test was used to compare the difference before and after flooding stress. For correlation analyses, Pearson’s correlation coefficient was calculated to determine the association of wilting caused by flooding damage and other photosynthesis parameters. *p*-values of less than 0.05 were considered statistically significant.

## Figures and Tables

**Figure 1 plants-11-00249-f001:**
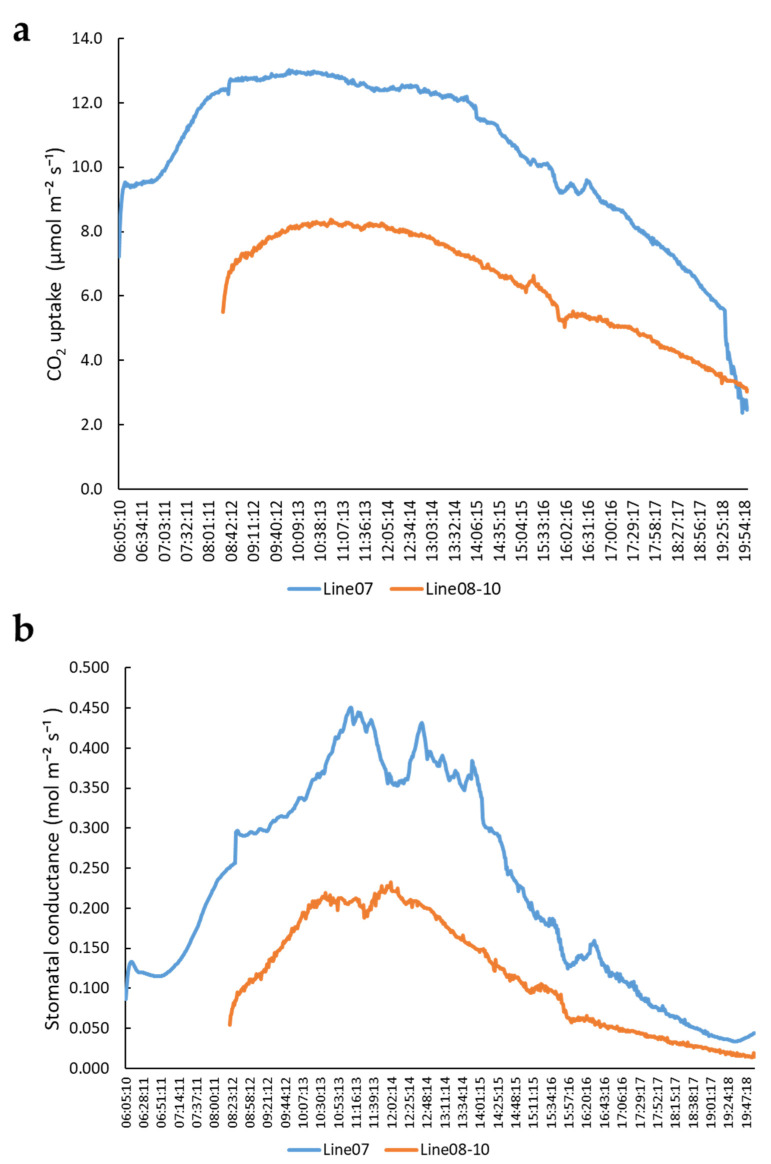
The variation of CO_2_ uptake and stomatal conductance of sacha inchi seedlings: (**a**) CO_2_ uptake and (**b**) stomatal conductance of sacha inchi Line 07 and Line 08–10 seedlings. Li-Cor chamber conditions were: temperature, 32 °C; light intensity, 1000 µmol m^−2^s^−1^; CO_2_, 400 µmol mol^−1^; and RH, 60%. An auto logging program was used to record data every 60 s. Line 07 was recorded from 06:00 to 19:50 but Line 08-10 was recorded from 08:15 to 19:50.

**Figure 2 plants-11-00249-f002:**
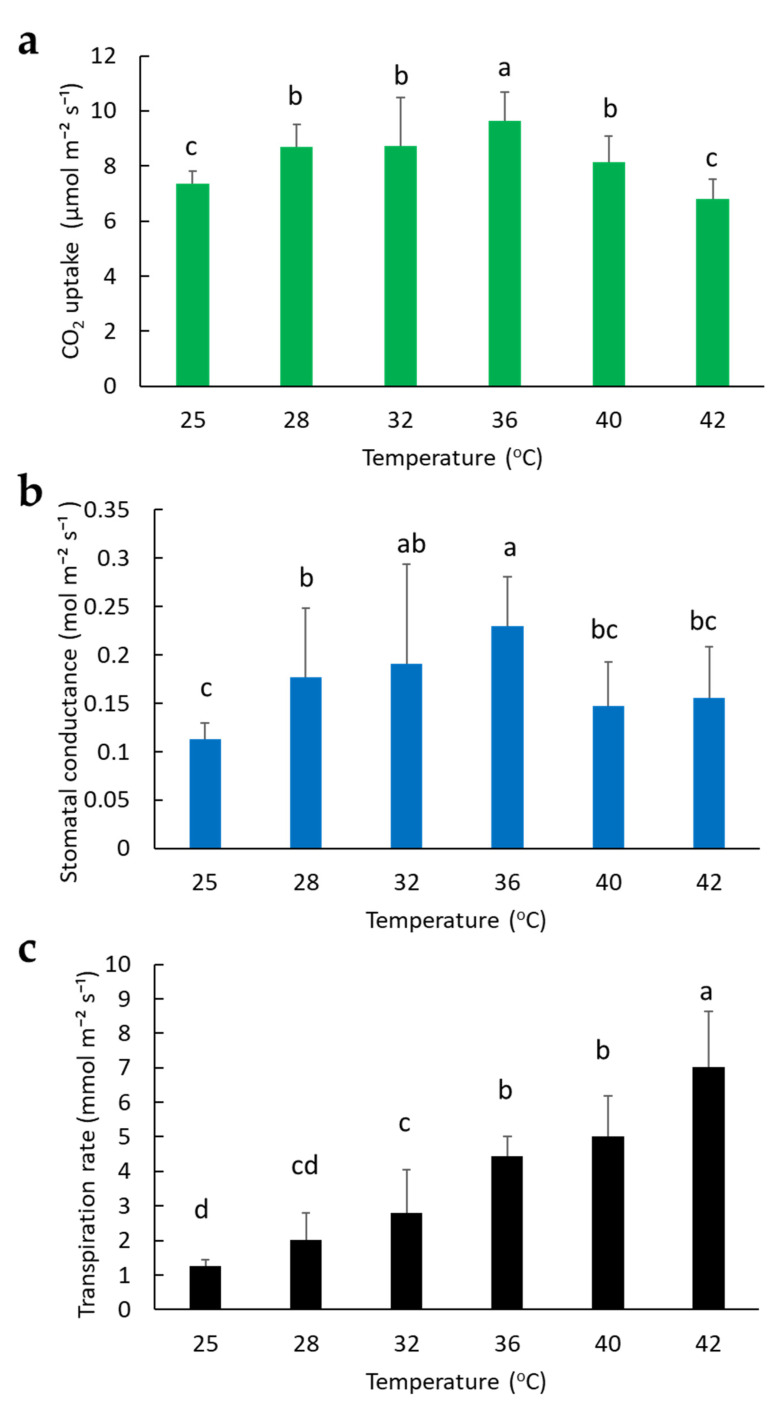
Effect of temperature on photosynthesis rate of sacha inchi: (**a**) CO_2_ uptake, (**b**) stomatal conductance, and (**c**) transpiration rate of sacha inchi seedling Line 07. In the Li-Cor chamber, the light intensity was set at 1000 µmol m^−2^s^−1^, RH to 60%, with 400 µmol mol^−1^ CO_2_, and temperatures were varied. Photosynthesis measurements were taken between 09:00 to 15:00. Statistical significance was determined by a one-way analysis of variance (ANOVA), followed by Duncan’s multiple range test (DMRT). Values followed by a different letter(s) are significantly different at *p* ≤ 0.05. Error bars indicate the SD of the mean from six seedlings.

**Figure 3 plants-11-00249-f003:**
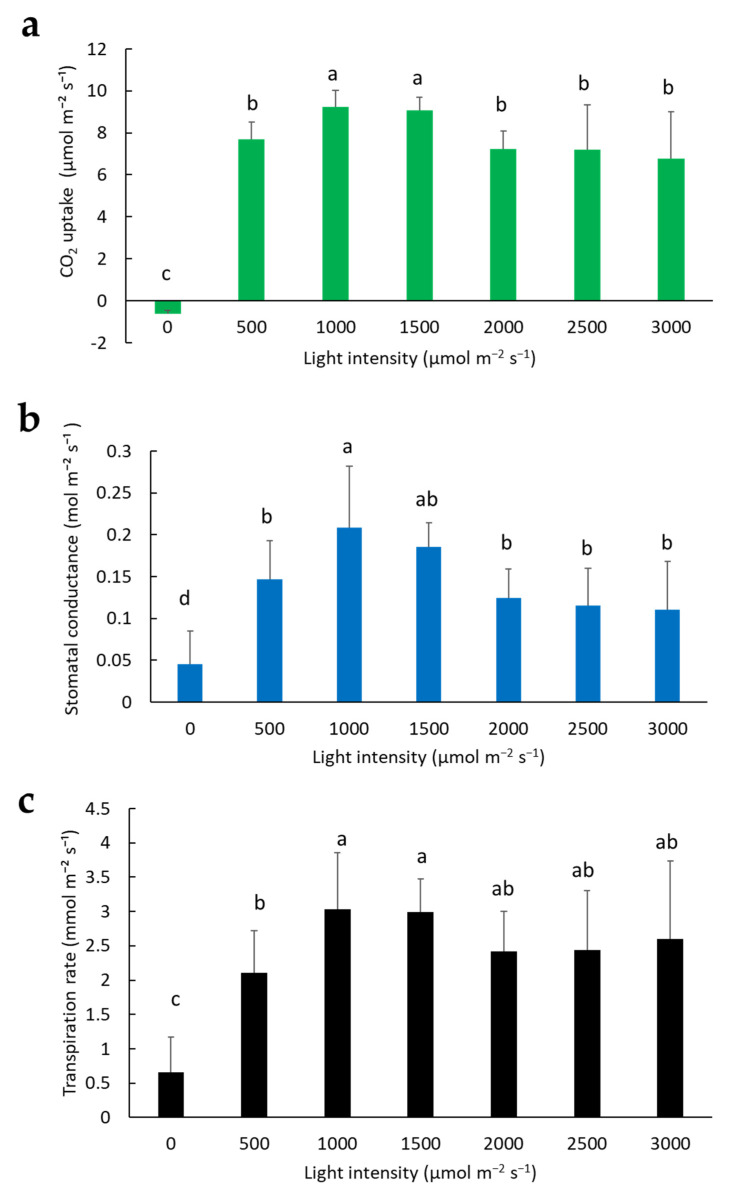
Effect of light intensities on photosynthesis rate of sacha inchi: (**a**) CO_2_ uptake, (**b**) stomatal conductance, and (**c**) transpiration rate of sacha inchi seedling Line 07. The Li-Cor chamber was set at a constant 32 °C, 60% RH, and 400 µmol mol^−1^ CO_2_, and varying light intensities were tested. Photosynthesis measurements were taken between 09:00 to 15:00. Statistical significance was determined by ANOVA. Values followed by a different letter(s) are significantly different at *p* ≤ 0.05 according to the DMRT test. Error bars indicate the SD of the mean from six seedlings.

**Figure 4 plants-11-00249-f004:**
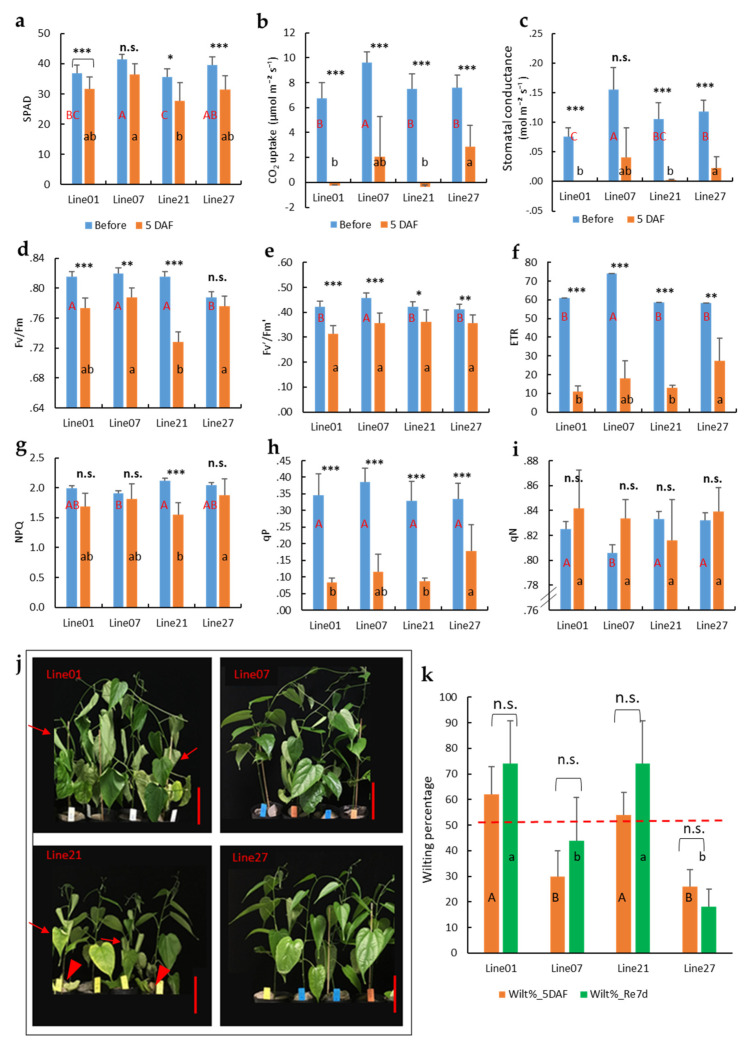
Response of different sacha inchi genotypes to waterlogging: (**a**) SPAD, (**b**) CO_2_ uptake, (**c**) stomatal conductance, (**d**–**i**) PSII parameters, (**j**) morphology of seedlings at 5 DAF, (**k**) leaf witling percentage at 5 DAF and witling percentage after recovery for 7 days in the greenhouse. Arrows show the wilting leaves; arrowheads indicate the fallen leaves after waterlogging. Bars, 10 cm. Sacha inchi seedlings of Lines 01, 07, 21, and 27 were waterlogged under a water depth of 5 cm above the soil surface. Photosynthesis measurements were taken between 09:00 to 15:00. Fv/Fm, the maximum quantum efficiency of the photosystem II; ETR, electron transport rate; NPQ, nonphotochemical quenching; qP, photochemical quenching of fluorescence, and qN, nonphotochemical quenching of fluorescence. Statistical significance was determined by one-way ANOVA, followed by Duncan’s multiple range test. Different capital letters indicate significant differences among genotypes before waterlogging by DMRT (*p* < 0.05). Different lowercase letters indicate significant differences among genotypes at 5 DAF by DMRT (*p* < 0.05). The student’s *t*-test was used to find the significant difference before and after waterlogging for 5 days. * *p* < 0.05, ** *p* < 0.01, *** *p* < 0.001, and n.s., no significant difference. Error bars represent the standard error of the mean (*n* = 6).

**Figure 5 plants-11-00249-f005:**
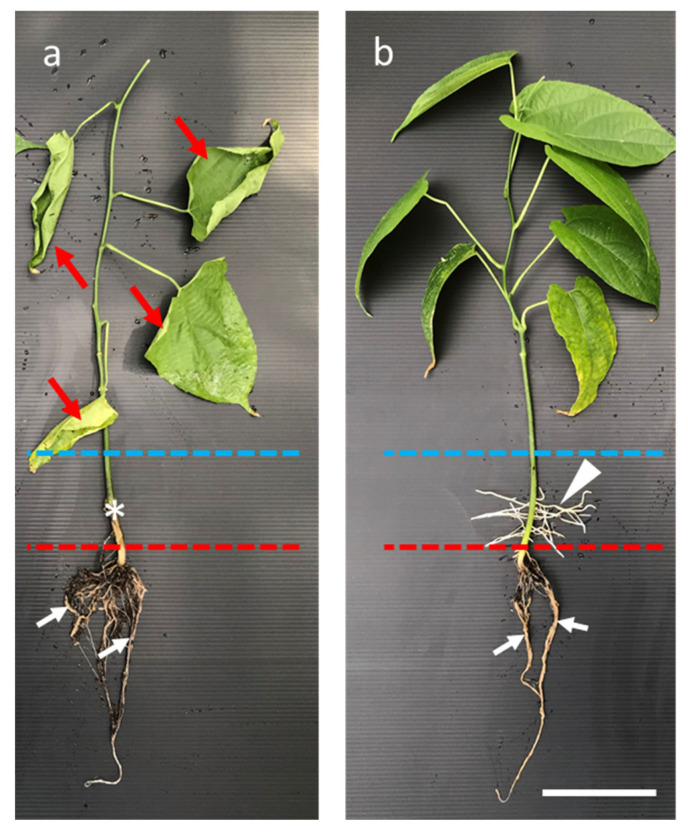
Morphology of sacha inchi seedlings after waterlogging treatment for two weeks: (**a**), the susceptible seedling has wilting leaves (red arrowheads), and the primary and lateral roots (arrows) and the submerged hypocotyl (asterisk, *) were rotten. (**b**), the tolerant seedling produced aquatic adventitious roots in the submerged hypocotyl region (arrowhead). Seedlings were waterlogged at 5 cm above the soil surface for two weeks. Blue dashed lines indicate the water surface and red dashed lines indicate the soil surface. Bar, 5 cm.

**Table 1 plants-11-00249-t001:** Pearson correlation coefficients of sacha inchi wilting percentage, chlorophyll content, gas exchange, and photosystem II traits at five days after waterlogging.

Trait	SPAD at 5 DAF	Wilt % at 5 DAF
Wilt % at 5 DAF	−0.413 *	1.00
Stomatal conductance	0.436 *	−0.670 **
CO_2_ uptake	0.458 *	−0.741 **
Transpiration rate	0.425 *	−0.692 **
Fv/Fm	0.746 **	−0.545 **
Fv′/Fm′	0.210	−0.433 *
ETR	0.458 *	−0.713 **
NPQ	0.402	−0.613 **
qP	0.447 *	−0.710 **
qN	0.185	−0.207

DAF, days after waterlogging. *n* = 24 plants. * *p* < 0.05; ** *p* < 0.01.

## Data Availability

All data included in the main text.

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
