# Peer review of "Photosynthetic and Morphological Responses of Sacha Inchi (Plukenetia volubilis L.) to Waterlogging Stress"

_plants, 2022, doi:10.3390/plants11030249_

Round 1

Reviewer 1 Report

The article deals with a study on the response of Sacha Inchi to waterlogging stress. The aim of the study is interesting, however, the manuscript has serious flaws and missing data, so I recommend rejecting the manuscript in its current form.
Authors should consider comments below, among others, when they are improving the manuscript.

L114 Air temperature above 36 ºC also decreased gs
L116 Repeated
L158 Move to discussion
L161 symptoms such as.... did you evaluate those symptoms?
L165 Can gs values of 0.04 mol m-2 s-1 be considered moderately high? Similar values were recorded under no light conditions (Fig. 3c) and you considered that stomata were closed (L137).
L168. Do you consider that waterlogging at 10 cm depth did not lead to hypoxia? 
L168 Did you measure plant growth? It seems that you consider that waterlogging at 10 cm did not affect plant growth. Is it true?   
L182 Your title says effects on survival but you didn't include any data about it. How many plants did survive for each depth? Were the effects the same to all the four lines? In the next experiment you measure the wilting percentage at 5 DAF, did you measure the wilting percentage in this experiment?
L185 You didn't compare the different genotypes after waterlogging for 5 days. It seems that you compared to each genotype the initial and the final situation.
L187 Please, define severe damage. 
L212 You did a Pearson correlation analysis between SPAD and the wilting percentage. You showed SPAD values in Fig. 5a. However, you did not show the wilting percentage for each line. You must include the wilting percentage at %DAF and after the 7 days recovery.
L233-L245 Repeated! L110 
L266 Can this sentence be applied to your experiment when the difference between the depth of both treatments is 10 cm? 
L285 Regarding your discussion, it is really interesting for farmers to know if any of the assessed lines are resistant to waterlogging. In my opinion, you need to show these results and discuss them. Section "2.4 Sacha Inchi survival" needs to be improved.
L289 Repeated! L277
L295 What about Line07? Are there significant differences between Line 07 and Line27? Can be both considered flood tolerant? In M&M you consider Line 07 "Good yielding and flooding tolerant" (L342)
L304 We also observed that some flooding tolerant lines produced AR after 10 DAF. Where can I see that? These values aren't included in the current version of the manuscript. If they are already published, please add the reference. If not, include them in the results section as it is a really interesting effect of waterlogging which can be an important factor to explain the differences reported among lines in gas exchange.
L305 Some lines? Which lines? Say the name!
L311 Repeated! L233
L333 When did you use Lines 8-10? You previously stated that the diurnal changes of photosynthesis rate were measured in Line07 (L103 - Figure 1).
L377 You must show the results related to the wilting percentage. 

If we compare the gs values reported in Fig. 1 for the Line 07 at 32ºC, light intensity 1 mmol m-2 s-1, CO2 0.4 mmol mol-1 and RH 60 % during the period from 9 am to 3 pm, all the values reported are higher than 0.27 mol m-2 s-1. However in Fig 2 and 3 the mean value provided is lower than 0.22 mol m-2 s-1. And in Fig 5c the gs mean value reported for the Line 07, before the experiment, is 0.15 mol m-2 s-1.  How can you explain that?
In the caption of the figures 2, 3 and 4 you should state that measurements were taken between 09.00 to 15.00h. 
In Fig. 4 you must include the 4 seedlings.
In Fig. 5, it seems that you are comparing to each line the values before (blue) and after the waterlogging (orange). You drew a bracket in fig. 5a Line01. You must clearly show if there were differences among the lines before and after the experiment.
In Fig. 5, you calculated the SEM and in the same cases it seems that the error bars do overlap but you state that there are significant differences. Fig 5a, Fig 5e Line 21, Fig. 5i Line 01.
In Fig. 5c, please check the vertical axis. 
Table 1. Please be consistent, neither photosynthesis rate nor the abbreviation "A" were used during the manuscript. Why did you decide to use them in the table? 
Why did you choose 5 days for the experiment shown in Fig 5 and 6 days for the one in Fig. 4?
Why did you choose 6 days of flooding? Is it a normal period in your area? In L283 you talk about heavy rains in the summer season, were you trying to simulate those conditions?
You should discuss the effects of waterlogging depth regarding the root distribution. Are there differences among the seedlings?

Author Response

Reviewer#1

The article deals with a study on the response of Sacha Inchi to waterlogging stress. The aim of the study is interesting, however, the manuscript has serious flaws and missing data, so I recommend rejecting the manuscript in its current form.
Authors should consider comments below, among others, when they are improving the manuscript.

Response: I sincerely appreciate your time, careful reading, and detail corrections of our paper. All suggestions were taking care and revised accordingly for the improvement of this article. Hope it is OK. Thank you 

L114 Air temperature above 36 ºC also decreased gs

Response: The sentence had been changed to “These data indicated that sacha inchi seedlings increased stomatal opening and have better photosynthesis capacity at 32°C and 36°C. Temperatures > 36°C increases transpiration rate that might lead to severe water loss. (Line 122).   
L116 Repeated

Response: We had deleted the sentence
L158 Move to discussion

Response: OK, the sentence had been moved to Line 217.
L161 symptoms such as.... did you evaluate those symptoms?

Response: We added wilting percentage data to compare water 10 cm and 20 cm (Figure 5a).  
L165 Can gs values of 0.04 mol m-2 s-1 be considered moderately high? Similar values were recorded under no light conditions (Fig. 3c) and you considered that stomata were closed (L137).

Response: Thanks for pointing this out. The sentence has been changed to “However, seedlings waterlogged in a depth of 10 cm maintained relatively high CO2 uptake and stomatal conductance compared to in a depth of 20cm (Figure 5b, c).” (Line 221-223)

L168. Do you consider that waterlogging at 10 cm depth did not lead to hypoxia? 

Response: No, waterlogging for 6 days even at 10cm also lead to hypoxia and show some level of wilting (Fig.5a). We revised the sentence to “These results indicate that waterlogging in a 20 cm deep water caused more severe hypoxia than in 10 cm water and significantly affects sacha inchi photosynthesis ability and growth.”(Line 226-228)

L168 Did you measure plant growth? It seems that you consider that waterlogging at 10 cm did not affect plant growth. Is it true?   

Response: No we didn’t measure growth data. Waterlogging at 10 cm water caused >30% wilting (Figure 5a).  
L182 Your title says effects on survival but you didn't include any data about it. How many plants did survive for each depth? Were the effects the same to all the four lines? In the next experiment you measure the wilting percentage at 5 DAF, did you measure the wilting percentage in this experiment?

Response: The title had been revised to "2.6. Effect of water depth on photosynthesis rate of sacha inchi "(Line 215)

L185 You didn't compare the different genotypes after waterlogging for 5 days. It seems that you compared to each genotype the initial and the final situation.

Response: We performed DMRT to compare different genotypes before and after waterlogging, respectively in this version. The figure 4 looks quite busy but provide more information. Thanks.  
L187 Please, define severe damage. 

Response: This sentence was deleted. In this revision, we showed data of four genotype (Figure 4) before water depths (Figure 5).
L212 You did a Pearson correlation analysis between SPAD and the wilting percentage. You showed SPAD values in Fig. 5a. However, you did not show the wilting percentage for each line. You must include the wilting percentage at %DAF and after the 7 days recovery.

Response: Wilting percentage data at 5 DAF and after 7 days recovery were included in Figure 4k and i. Thanks.
L233-L245 Repeated! L110 

Response: The sentences were modified.
L266 Can this sentence be applied to your experiment when the difference between the depth of both treatments is 10 cm? 

Response: Yes I think so, although just a 10 cm different of water depth but after prolong waterlogging for several days we could see significant difference in Line07.
L285 Regarding your discussion, it is really interesting for farmers to know if any of the assessed lines are resistant to waterlogging. In my opinion, you need to show these results and discuss them. Section "2.4 Sacha Inchi survival" needs to be improved.

Response: OK, we changed the title to “Effect of water depth on photosynthesis rate of sacha inchi”.
L289 Repeated! L277

Response: Line 289 sentence was deleted.
L295 What about Line07? Are there significant differences between Line 07 and Line27? Can be both considered flood tolerant? In M&M you consider Line 07 "Good yielding and flooding tolerant" (L342)

Response: Thanks. Yes, Line07 and 27 showed no significant difference after waterlogging (Figure 4). So both could be grouped as tolerant lines.  
L304 We also observed that some flooding tolerant lines produced AR after 10 DAF. Where can I see that? These values aren't included in the current version of the manuscript. If they are already published, please add the reference. If not, include them in the results section as it is a really interesting effect of waterlogging which can be an important factor to explain the differences reported among lines in gas exchange.

Response: In fact, this is the first paper of sacha inchi in my Lab. We observed some seedlings produce AR can survive after waterlogging. We will carry out detail study in this area in the near future.
L305 Some lines? Which lines? Say the name!

Response: Lines 07, 27
L311 Repeated! L233

Response: The first two sentences had been deleted.
L333 When did you use Lines 8-10? You previously stated that the diurnal changes of photosynthesis rate were measured in Line07 (L103 - Figure 1).

Response: We had measured Line 8-10 and Line07 for the diurnal changes of photosynthesis rate. Both have similar trends. We just presented data of Line07 because this line was used to study temperature and light effect on photosynthesis. We explained more in this version “we used a Li-Cor 6800 to detect CO2 uptake and stomatal conductance of two lines (Line 8-10 and Line 07) under constant 1000 µmol m2s1 irradiation to avoid sunlight fluctuation in the greenhouse. The results indicated that both lines have similar diurnal photosynthesis curves. The data for Line 07 are reported in Figure 1. ” (Line94-98).
L377 You must show the results related to the wilting percentage. 

Response: Thanks for the reminder. We included wilting percentage data of 5DAF (Figure 4k) and recovery 7 days (Figure 4l) in this version.  

If we compare the gs values reported in Fig. 1 for the Line 07 at 32ºC, light intensity 1 mmol m-2 s-1, CO2 0.4 mmol mol-1 and RH 60 % during the period from 9 am to 3 pm, all the values reported are higher than 0.27 mol m-2 s-1. However in Fig 2 and 3 the mean value provided is lower than 0.22 mol m-2 s-1. And in Fig 5c the gs mean value reported for the Line 07, before the experiment, is 0.15 mol m-2 s-1.  How can you explain that?
In the caption of the figures 2, 3 and 4 you should state that measurements were taken between 09.00 to 15.00h. 

Response: The variations of gs values might due to different batches of experiment and environmental factors in the greenhouse. Normally, fluctuation of stomatal conductance is greater than CO2 uptake. We had mentioned photosynthesis measurements were taken between 09:00 to 15:00h. Thank you.

In Fig. 4 you must include the 4 seedlings.

Response: Sorry that we take chlorophyll fluorescence data in a “dark room” to obtain Fv/Fm. Some seedlings were broken during measurement. Therefore we didn’t take seedlings photo.  
In Fig. 5, it seems that you are comparing to each line the values before (blue) and after the waterlogging (orange). You drew a bracket in fig. 5a Line01. You must clearly show if there were differences among the lines before and after the experiment.

Response: Thanks for the good suggestion. We carried out statically analysis and compared four genotypes before and after waterlogging in this version and marked the differences in the same Figure.   
In Fig. 5, you calculated the SEM and in the same cases it seems that the error bars do overlap but you state that there are significant differences. Fig 5a, Fig 5e Line 21, Fig. 5i Line 01.

Response: Thanks for your careful in reading. We checked again our statitic data and found one mistake. Fig. 5a Line21 had been corrected to one *. The other two are correct. FYI.
In Fig. 5c, please check the vertical axis. 

Response: Ok, the number had been revised. Thanks.
Table 1. Please be consistent, neither photosynthesis rate nor the abbreviation "A" were used during the manuscript. Why did you decide to use them in the table? 

Response: Ok, we changed A to CO2 uptake. Thanks
Why did you choose 5 days for the experiment shown in Fig 5 and 6 days for the one in Fig. 4?

Response: We decide to measure photosynthesis at 5 DAF mainly due to Line01 and Line21 were sensitive to waterlogging and showed severe wilting. We waterlogged Line07 for 6 days for in two water depths mainly because Line07 is more flood tolerance.  

Why did you choose 6 days of flooding? Is it a normal period in your area? In L283 you talk about heavy rains in the summer season, were you trying to simulate those conditions?

Response: Yes. long duration of rain is common in Taiwan, especially during typhoon or monsoon season.

You should discuss the effects of waterlogging depth regarding the root distribution. Are there differences among the seedlings

Response: Sorry we did not study root in this study. Basically, the root growth should have some variations among genotype. We are interested to study AR development in future. FYI.  

Reviewer 2 Report

The MS focused on waterlogging stress tolerance in Sacha Inchi (Plukenetia volubilis L.) as it showed from the title. However, I am unable to understand the use of various temperature and light intensities. Whether the authors tend to assess changes in flooding stress tolerance with the change in diurnal light intensities and temperature? If so, this has not been described as main objectives in Introduction section.  However, all sections of the MS are poorly written. The experimental design is again poor. Flooding stress or hypoxic conditions of various degree should be compared with normoxic or normal oxygen conditions or Control conditions. In this situation, Control, Hypoxic conditions by applying irrigation/flooding up to 10 cm depth and flooding up to 20 cm depth. In addition, how we can assess whether plant is experiencing hypoxic conditions or not. For this purpose, it is suggested to add data for soil redox potential. over 4 days. Or present soil nutrient data particularly that of NO3-N, NH4-N, Fe and Mn which reflect the soil redox potential or soil reactions. 
Abstract Do not reflect the essence of studies or findings
Introduction: Please add the literature how flooding stress affect soil oxygen, soil reaction, which affect the gas exchange, and Photosystem II activity. Cite different responses of tolerant and sensitive species/cultivars in terms of gas exchange, and Photosystem-II activities 

Results: Description needs improvement
Discussion section is poorly written It needs to be revised.  
M&M. Design of the experiment is poorly described. This section needs revisions.
Add Soil redox potential data over time or any other supportive data which reflect the intensity of flooding stress that which were experience by the plants. 

Author Response

Reviewer#2

The MS focused on waterlogging stress tolerance in Sacha Inchi (Plukenetia volubilis L.) as it showed from the title. However, I am unable to understand the use of various temperature and light intensities. Whether the authors tend to assess changes in flooding stress tolerance with the change in diurnal light intensities and temperature? If so, this has not been described as main objectives in Introduction section.  However, all sections of the MS are poorly written. The experimental design is again poor. Flooding stress or hypoxic conditions of various degree should be compared with normoxic or normal oxygen conditions or Control conditions. In this situation, Control, Hypoxic conditions by applying irrigation/flooding up to 10 cm depth and flooding up to 20 cm depth. In addition, how we can assess whether plant is experiencing hypoxic conditions or not. For this purpose, it is suggested to add data for soil redox potential. over 4 days. Or present soil nutrient data particularly that of NO3-N, NH4-N, Fe and Mn which reflect the soil redox potential or soil reactions. 
Response: Thanks for your comments. This paper is submitted to a special issue on “Photosynthesis under Environmental Fluctuations”. There are two main objectives of this study, (1) to characterize sacha inchi photosynthesis behavior (daily change, temperatures and light intensities; and (2) to study it physiological changes under waterlogging stress. We had changed our title to “Sacha Inchi (Plukenetia volubilis L.) Photosynthesis Behavior and Response to Waterlogging Stress”. 

In this revised manuscript, we have cited reference of soil redox in introduction (Line 61~64) “In well-aerated field conditions, soil redox (Eh) ranges from 400~700 mV, which favors oxidation and aerobic microorganism growth. However, waterlogging decreases Eh, reduces the soil oxygen concentration, and represses soil microhabitats [12].”. 

Abstract Do not reflect the essence of studies or findings

Response: We have revised Abstract in this revised version. Hope it is OK.    

Introduction: Please add the literature how flooding stress affect soil oxygen, soil reaction, which affect the gas exchange, and Photosystem II activity. Cite different responses of tolerant and sensitive species/cultivars in terms of gas exchange, and Photosystem-II activities.  

Response: Thanks for the suggestions. We had cited some relevant references in introduction.  

Results: Description needs improvement

Response: We had improved.   
Discussion section is poorly written. It needs to be revised.

Response: We have made revisions as much as we can and this revised version had been edited by MDPI for the English editing service.

M&M. Design of the experiment is poorly described. This section needs revisions.
Add Soil redox potential data over time or any other supportive data which reflect the intensity of flooding stress that which were experienced by the plants. 

Response: We have improved the M&M section. Thanks for the suggestion on measuring soil redox potential during flooding. In this study, sacha inchi seedlings were planted in a pot containing soilless peak moss. We agree that soil redox is an important indicator during flooding in the field. We will measure redox for the large-scale flooding experiment in the field.   

Reviewer 3 Report

Your work about the response of Sacha Inchi to waterlogging gives some new information about this plant and its cultivation. I do not have many important advices to give to you for improving the quality of your experiment but I suggest to have an English revision. The repetition of "Sacha inchi" is sometime really exagerate such as in rows 44-52 where you repeat the words 5 times. You can use: species, plants, plukenetia... or change the sentences to avoid multiple repetition.  The sentence at rows 71-74 seems just a "cut and paste" you would better write this together with rows 74-78. 

The sentence at rows 291-292 is not clear. Did  you mean "seedlings from different genetic lines?" 

In future prospective line 319-322 are not very clear to me. You mean that the the research on this crop cold also take advantages from the development of methods for in vitro culture?

The response of the plants to light intensity also depends on the natural  environment they are adapted to. Could you please indicate the mean PAR or level of irradiation in the greenhouse?    

Author Response

Your work about the response of Sacha Inchi to waterlogging gives some new information about this plant and its cultivation. I do not have many important advices to give to you for improving the quality of your experiment but I suggest to have an English revision. The repetition of "Sacha inchi" is sometime really exagerate such as in rows 44-52 where you repeat the words 5 times. You can use: species, plants, plukenetia... or change the sentences to avoid multiple repetition.  The sentence at rows 71-74 seems just a "cut and paste" you would better write this together with rows 74-78. 

Response: Thanks for your kind suggestion. We had revised accordingly.  

The sentence at rows 291-292 is not clear. Did you mean "seedlings from different genetic lines?" 

Response: It is changed to “Maintenance of a high photosynthesis rate is crucial for flooding tolerance in plants. In response to flooding, tolerant genotypes tend to not show significant changes in net photosynthesis rate and slightly decreased stomatal conductance [19].” (Line 291~293).

In future prospective line 319-322 are not very clear to me. You mean that the research on this crop cold also take advantages from the development of methods for in vitro culture?

Response: Thanks for the comment. We delete the sentence in this version.  

The response of the plants to light intensity also depends on the natural environment they are adapted to. Could you please indicate the mean PAR or level of irradiation in the greenhouse?

Response: We had mentioned PAR in the greenhouse. Please refer to Lines 366~377. Thanks.

Reviewer 4 Report

TITLE

You also studied photosynthesis response to light and temperature but your title does not say this. Your title should cover at least the majority of your experiments.

INTRODUCTION

Pag 1, line 37. Replace “antioxidant activities” with “antioxidants”.

Pag 1, line 38. “huge” is not a formal word to be used in scientific articles.

Lines 44-51. What do you mean by “gas exchange parameters”? It can mean a lot of things, even photosynthesis itself. Be specific about the “severe stress” caused at temperatures below 8°C. What are the maximum photosynthetic rates of this species? What are the PPFD at maximum photosynthesis? Saying “full light” is not enough for us to know what is the optimum light range for this species. What is the PPFD of “100% light”?

Lines 54-55. Are you talking about submergence or soil waterlogging? I believe that flooding can be either of them so you need to be more specific. Your title says soil waterlogging and the effects on plants are very different from plant submergence. Be more specific. Besides, sacha inchi is a vine and it is unlikely to suffer from complete submergence unless when they are very young.

Lines 57-58. “Flooding stress causes rapid stomatal closure due to abscisic acid (ABA) and ethylene release, and hence reduced photosynthesis [11].” I am not sure whether this is consensus. I think the causes for stomatal closure during flooding are still under debate. Again, make sure you are clearer about whether you are referring to soil waterlogging or submergence.

Lines 80-81. “However, there has been no detailed study on photosynthesis behavior and how the plant reacts to flooding stress conditions.”. What is the importance of studying flooding in Sacha inchi? Does this species suffer from flooding? How is the cultivation of this species? Do people grow this species for agricultural purposes?

Overall, the introduction needs to be improved. Sentences needs to be restructured. The English needs a lot of improvement. Also, what are the hypothesis?

METHODS

Overall, methods lack several important information that are very important for the paper.

Line 332. Please provide where the farms where you collected seeds were located, geographic coordinate, climate conditions, and how old were the plants. From how many plants did you collect seeds? Did you bring them to the lab immediately? Did you store them?

Line 335. What is RO water?

Line 336. Did you germinate seeds in a growth chamber or in the greenhouse? At what light condition? Why 32°C? Was it according this article (https://www.scielo.br/j/jss/a/sQgzwrwhwgC3V979WGXhXmQ/?lang=en)?

Line 337. What was the volume of the pots? What were the light conditions of the greenhouse? Did you water them every day? Did you fertilize the plants? How old were they when you started the experiments? What size?

Line 341. “We used a good agronomy trait Sacha Inchi ‘Lines 8-10’”. What does that mean? What is a good agronomy trait? What does the lines represent? Different genotypes? How were they produced? By plant breeders?

Line 343. Was each seedling in separated pots?

Lines 345-346. “…performed flooding at 345 two different water depths, 10 cm and 20 cm, from the bottom of the pot.” Did these conditions cover the whole pot or only part of the pots? How many cm above the soil surface? Did you watered the plants in order to maintain the water column always in the same place? Did you cover the pots to avoid algae proliferation and thus water reoxygenation? Do you have pictures to illustrate the plants and the experiment? What did you consider a Bio-replicate? Each seedling? Were the four seedlings inside the same aquarium? Why 5 days of flooding? Was it enough to cause stress?

Line 329. “Each line had six seedlings.” Six repetitions per line and per treatment or four? I did not understand.

Line 353. Replace “CO2 uptake” by “photosynthesis”. They do not mean the same thing. Sometimes CO2 can enter the leaf but not be used for photosynthesis.

Line 355. Why did you perform experiments from 9:00 to 15:00? Your diurnal experiment says photosynthesis peak from 09:00 to 13:00. By the way, you forgot to mention how you collected data for this experiment in the material and methods.

Line 359. What temperature did you use in the study of light effect on photosynthesis?

RESULTS AND FIGURES

Fig 1. Line 105. Did you use an artificial light of 1000 to measure how photosynthesis vary within the day? How many plants did you measure? Was there natural light at 19:00?

Fig 2. Line 121. “Effect of temperature on photosynthesis rate of Sacha Inchi”. You should mention that this was at short-term test. You did not grow plants at different temperatures, you only changed the LICOR temperature.

Figures 2-3. Transform transpiration data from mol to mmol. It is more common this way.

Line 157. Effect of waterlogging depth on Sacha Inchi. Didn’t you have control plants? You should always have control plants when studying a stress.

Fig. 5. Line 203. “(j) Morphology of seedlings at 5 DAF’. These plants look great. They did not show any sign of damage or stress. Were they at 10 or 20 cm of waterlogging?

Author Response

Reviewer#4

You also studied photosynthesis response to light and temperature but your title does not say this. Your title should cover at least the majority of your experiments.

Response: We change our title to “Sacha Inchi (Plukenetia volubilis L.) Photosynthesis Behavior and Response to Waterlogging Stress”, hope it is better.

INTRODUCTION

Pag 1, line 37. Replace “antioxidant activities” with “antioxidants”.

Response: OK, it had been changed.

Pag 1, line 38. “huge” is not a formal word to be used in scientific articles.

Response: It is replaced to “enormous” 

Lines 44-51. What do you mean by “gas exchange parameters”? It can mean a lot of things, even photosynthesis itself. Be specific about the “severe stress” caused at temperatures below 8°C. What are the maximum photosynthetic rates of this species? What are the PPFD at maximum photosynthesis? Saying “full light” is not enough for us to know what is the optimum light range for this species. What is the PPFD of “100% light”?

Response: “gas exchange parameters” has been changed to “stomatal conductance, transpiration rate”. “severe stress” has been changed to “cold stress”. The reduction of Fv/Fm in the three species were included in this revised manuscript. The paper of Cai, 2011 didn’t mentioned about the PPFD of 100% light. In fact, natural sunlight is fluctuation. FYI.    

Lines 54-55. Are you talking about submergence or soil waterlogging? I believe that flooding can be either of them so you need to be more specific. Your title says soil waterlogging and the effects on plants are very different from plant submergence. Be more specific. Besides, sacha inchi is a vine and it is unlikely to suffer from complete submergence unless when they are very young.

Response:  Thanks for the kind reminder. Our study used waterlogging, not submergence. To be more specific, we had changed flooding to waterlogging in the text.    

Lines 57-58. “Flooding stress causes rapid stomatal closure due to abscisic acid (ABA) and ethylene release, and hence reduced photosynthesis [11].” I am not sure whether this is consensus. I think the causes for stomatal closure during flooding are still under debate. Again, make sure you are clearer about whether you are referring to soil waterlogging or submergence.

Response:  I agree different plant species may have different response against flooding stress. Flooding was change to ‘waterlogging’ in this revised version. Thanks

Lines 80-81. “However, there has been no detailed study on photosynthesis behavior and how the plant reacts to flooding stress conditions.”. What is the importance of studying flooding in Sacha inchi? Does this species suffer from flooding? How is the cultivation of this species? Do people grow this species for agricultural purposes?

Response:  Sacha inchi is planted widely in tropical rainforest. Heavy rain or typhoon season caused production loss. We added the increasing cultivation in Taiwan recently due to its economic value. Please refer to Line 254-255 “Sacha inchi is an oil crop with considerable potential and high economic value [3]. Taiwan just started planting this crop in 2015, and the planting area now exceeds 1200 hectares. “.   

Overall, the introduction needs to be improved. Sentences needs to be restructured. The English needs a lot of improvement. Also, what are the hypothesis?

Response:  Thanks for the suggestion. This revised version was submitted to MDPI for English editing for the readable of this article. “We hypothesized that the high photosynthesis capacity may help sacha inchi to withstand waterlogging stress. “ (Line 88~89). Thank you.   

METHODS

Overall, methods lack several important information that are very important for the paper.

Response:  Thanks for the reminder. We have include more detail information in this revision.

Line 332. Please provide where the farms where you collected seeds were located, geographic coordinate, climate conditions, and how old were the plants. From how many plants did you collect seeds? Did you bring them to the lab immediately? Did you store them?

Response:  We included more information related to this study in M&M section 4.1. (Line 346)  

Line 335. What is RO water?

Response:  Reverse osmosis (RO)

Line 336. Did you germinate seeds in a growth chamber or in the greenhouse? At what light condition? Why 32°C? Was it according this article (https://www.scielo.br/j/jss/a/sQgzwrwhwgC3V979WGXhXmQ/?lang=en)?

Response:  Thanks for providing the web information. We had very good seed germination rate >90% for sacha inchi. We had performed seed germination test at 28 to 35oC and found that 32 oC has the highest germination rate, second high at 30 oC, less germination at <28 oC or > 35 oC under dark conditions. So, we germinated seed at 32 oC in an incubator.

Line 337. What was the volume of the pots? What were the light conditions of the greenhouse? Did you water them every day? Did you fertilize the plants? How old were they when you started the experiments? What size?

Response: Pot volume is 7.5 cm width × 7.5 cm height, 140 mL volume (Line 342). We irrigated based on the need for seedlings. More irrigation during hot summer but less if it was cloudy.  We will water carefully to avoid waterlogging. There was no need to fertilize 1.5 months after sowing. We added, “Sacha inchi seedlings at the five-leaf stage (approximately 40 cm height)” (Line 345). Thanks.   

Line 341. “We used a good agronomy trait Sacha Inchi ‘Lines 8-10’”. What does that mean? What is a good agronomy trait? What does the lines represent? Different genotypes? How were they produced? By plant breeders?

Response:  We had changed to “We used high-yielding sacha inchi Lines 8-10 and 07 (good yielding and tolerant to waterlogging) to study diurnal changes in photosynthesis. ” (Line 346~348). They are different genotypes but “To understand the daily changes in the photosynthesis of sacha inchi, we used a Li-Cor 6800 to detect CO2 uptake and stomatal conductance of two lines (Line 8-10 and Line 07) under constant 1000 µmol m-2s-1 irradiation to avoid sunlight fluctuation in the greenhouse. The results indicated that both lines have similar diurnal photosynthesis curves. The data for Line 07 are reported in Figure 1.” (Line 94-98)

Line 343. Was each seedling in separated pots?

Response: Yes, an individual seedling was planted in a separate pot.  

Lines 345-346. “…performed flooding at two different water depths, 10 cm and 20 cm, from the bottom of the pot.” Did these conditions cover the whole pot or only part of the pots? How many cm above the soil surface? Did you watered the plants in order to maintain the water column always in the same place? Did you cover the pots to avoid algae proliferation and thus water reoxygenation? Do you have pictures to illustrate the plants and the experiment? What did you consider a Bio-replicate? Each seedling? Were the four seedlings inside the same aquarium? Why 5 days of flooding? Was it enough to cause stress?

Response: for the waterlogging depths, we performed flooding at two different water depths, 10 cm, and 20 cm, from the bottom of the pot, which is 3 cm and 13 cm above peat moss surface. For the seedling waterlogged their leaves were above water. We added water to maintain the water depths at 10cm and 20cm (from the pot bottom). The seedlings were not submerging. We didn’t cover the pot and there were no algae observed during the six days of waterlogging. Four seedlings of both genotypes were placed in the same aquarium. We observed some wilting symptoms at 5 DAF and proceeded with photosynthesis measurement. Figure 5 showed a statically different between 10 and 20 cm. We included more detailed information in the M&M section. Thanks.      

Line 329. “Each line had six seedlings.” Six repetitions per line and per treatment or four? I did not understand.

Response: It was revised to “Six seedlings per line were placed in a plastic box (24 cm height × 60 cm length × 43 cm width) and waterlogged under a depth of 10 cm.” (Line 351~352)

Line 353. Replace “CO2 uptake” by “photosynthesis”. They do not mean the same thing. Sometimes CO2 can enter the leaf but not be used for photosynthesis.

Response:  Thanks, we had changed.

Line 355. Why did you perform experiments from 9:00 to 15:00? Your diurnal experiment says photosynthesis peaks from 09:00 to 13:00. By the way, you forgot to mention how you collected data for this experiment in the material and methods.

Response: We need more time to detect a large number and replicate samples. Moreover, each data take 10 minutes to be stabilized. Carried out Li-Cor measurement in the morning is not enough. Therefore, we perform Li-cor measurements from 09:00 to 15:00 before stomata were closed. Ok, it was added in the text. Thanks.

Line 359. What temperature did you use in the study of light effect on photosynthesis?

Response:  “ To study of the effect of light intensity on gas exchange, we set the chamber temperature at 32°C; and used seven levels of irradiation, i.e., 0, 500, 1000, 1500, 2000, 2500, and 3000 μmol photons m-2 s-1. We exposed newly established leaves to various temperature and light treatments for approximately 10 min until the CO2 uptake curve was stable, then data were recorded. “ (Line 374~379).

RESULTS AND FIGURES

Fig 1. Line 105. Did you use an artificial light of 1000 to measure how photosynthesis vary within the day? How many plants did you measure? Was there natural light at 19:00?

Response: Yes, we used LED light installed in Licor chamber, fix at 1000 PPFD to test the photosynthesis, stomatal conductance, and transpiration. Fig.3b data showed that stomatal opening is dependent on light, Sacha Inchi closed stomata under dark. To understand the diurnal change in photosynthesis in sacha inchi, we recorded gas exchange data from 06:00 (PAR in the greenhouse was <10 µmol m-2s-1) to 20:00 (PAR in the greenhouse was <0.2 µmol m m-2s-1). (Line 364~366)    

Fig 2. Line 121. “Effect of temperature on photosynthesis rate of Sacha Inchi”. You should mention that this was at short-term test. You did not grow plants at different temperatures, you only changed the LICOR temperature.

Response:  Thanks for the reminder. It has been included M&M (section 4.3). “We exposed newly established leave to various temperatures or lights conditions for approximately 10 minutes until CO2 uptake data was stable, then data was recorded”.

Figures 2-3. Transform transpiration data from mol to mmol. It is more common this way.

Response: Ok, the unit of transpiration had been changed to mmol m⁻² s⁻¹ and we have redrawn Fig. 2C and 3C. Thanks

Line 157. Effect of waterlogging depth on Sacha Inchi. Didn’t you have control plants? You should always have control plants when studying a stress.

Response: Thanks for pointing this out. Sorry that we didn’t include seedlings without waterlogging as a control when testing the effect of water depth mainly due to the lack of uniform seedlings. Moreover, without waterlogging, seedlings have good photosynthesis performance under normal growth conditions could be seen in the experiment of the four genotypes response to waterlogging. To make it more logical to read, we decide to place the response of genotypes against waterlogging (Figure 5) before the water depth exp’t (Figure 6) in this revised manuscript. Hope it is OK.

Fig. 5. Line 203. “(j) Morphology of seedlings at 5 DAF’. These plants look great. They did not show any sign of damage or stress. Were they at 10 or 20 cm of waterlogging?

Response:  At 5 DAF, some leaves were wilting and falling. We have added arrows and arrowheads in the figure to highlight the waterlogging symptoms. As mentioned in the figure legend of Fig. 5, “Seedlings were waterlogged in a depth of 10 cm of water.” Also, it was mentioned in the M&M. FYI, and Thanks. 

Round 2

Reviewer 1 Report

The manuscript has been improved by the authors. However, they have not yet answered several major questions, so my opinion remains the same. I cannot recommend the publication of the manuscript.

These are some points that had not been answered in the new version.

Figure 4 has been improved; however, it has not been properly discussed. There are several differences among seedlings before the waterlogging and after which deserve deep consideration and discussion.

You did not discuss the implication of those physiological changes. Seedling Line07 had the highest SPAD, CO2uptake and Gs before the experiment started but there were no significant differences 5DAF between Line07 and Line01, even Line07 is supposed to be a tolerant genotype and Line01 a vulnerable genotype. Moreover, there were no significantly differences in the % of wilting plants between both seedlings. Why is Line07 considered tolerant?

Gas exchange data measured after 7 days of recovery should be provided to understand Figure 4L.

Figure 4C hasn't been checked. As I already told you, the values in the vertical axis are repeated. Please write 2 decimal numbers and avoid repeating the same number twice.

L314 "Waterlogging-tolerant genotypes such as Lines 07 and 27 are able to adjust electron flow and maintain high photosynthesis under waterlogging conditions (Figure 4)." Line07 did not show significant differences with Line01 and Line21 after the waterlogging. If there are no significant differences, how can you consider that Line07 is able to adjust electron flow but Line01 is not? Moreover, why do you consider that values of 0.002 mmol m-2 s-1 CO2 uptakes are high?

You must critically discuss your results and compare them with other works. Figure 4 is poorly discussed in section 3.4. Moreover, you explain your results with data that you decide not to show (disease infection susceptibility and adventitious roots presence) If you consider that those factors are important to justify your results, you have to include them.

You did not answer/justify the disagreements found in your results. In Figure 5C, gs = 0.04 mol m-2 s-1 and you considered that it is a high stomatal conductance, while in Figure 3B you stated for a similar value that stomata were closed. 

L286 "Every increase of one meter in water depth reduces the gas saturation by 10%". You work in a range between 10 and 20 cm depth. You should also discuss papers that show similar conditions to those described in your experiment. Again, you justify your results according to an impaired root growth when you did not show any root development data. In my opinion, as you said that you are interested in studying Sacha inchi's root development, you should wait until you get your results to confirm or dispel your hypothesis.

Your answer to the question: if we compare the gs values reported in Figure 1 for the Line 07 at 32ºC, light intensity 1 mmol m-2 s-1, CO2 0.4 mmol mol-1 and RH 60 % during the period from 9 am to 3 pm, all the values reported are higher than 0.27 mol m-2 s-1. However, in Figures 2 and 3, the mean value provided is lower than 0.22 mol m-2 s-1. And in Figure 5c the gs mean value reported for Line 07, before the experiment, is 0.15 mol m-2 s-1. How can you explain that?

"The variations of gs values might due to different batches of experiment and environmental factors in the greenhouse. Normally, fluctuation of stomatal conductance is greater than CO2 uptake."

If you consider that the conditions in the greenhouse varied so much, you must show/discuss them. Which conditions did change and how did they change? Your gs values for the same seedling (Line07) and measuring conditions decreased by 40 %. 

Author Response

Reviewer#1

The manuscript has been improved by the authors. However, they have not yet answered several major questions, so my opinion remains the same. I cannot recommend the publication of the manuscript.

Response: Thanks for your time and constructive comments. We tried our best to revise accordingly to improve our manuscript. We apologize for the mistakes. We noted the discrepancy of the experiment of the water depth due to lack of control and not taking the photo of the seedlings. Therefore, we decide to delete this experiment. We replaced the new section 2.6 title toAdventitious root formation after long-term waterlogging. We provide some more information about AR contributing to flooding tolerance.

These are some points that had not been answered in the new version.

Figure 4 has been improved; however, it has not been properly discussed. There are several differences among seedlings before the waterlogging and after which deserve deep consideration and discussion.

Response: Thanks for pointing this out. We have revised the result and discussed more the finding of Figure 4 in the current version.

You did not discuss the implication of those physiological changes. Seedling Line07 had the highest SPAD, CO2 uptake, and Gs before the experiment started but there were no significant differences 5DAF between Line07 and Line01, even Line07 is supposed to be a tolerant genotype and Line01 a vulnerable genotype. Moreover, there were no significant differences in the % of wilting plants between both seedlings. Why is Line07 considered tolerant?

Response:  I apologize for the typo of “ab” in Line 01, it should be “a”. Its mean data is slightly higher than Line21 at 5 DAF.  Line07 was grouped to moderate tolerant in this study. “Overall, the order of waterlogging tolerance of these four genotypes is: Line27 > Line07 > Line01 = Line21.” (Line 282).

Gas exchange data measured after 7 days of recovery should be provided to understand Figure 4L.

Response:  Sorry, we did not measure the gas exchange data after recovery. To avoid the miss understanding, we had deleted the recovery 7 days data from Table 1 (Pearson correlation).   

Figure 4C hasn't been checked. As I already told you, the values in the vertical axis are repeated. Please write 2 decimal numbers and avoid repeating the same number twice.

Response: Sorry, we haven’t noticed the mistake! It had been revised to 2 decimal numbers. Thanks for your watchfulness.  

L314 "Waterlogging-tolerant genotypes such as Lines 07 and 27 are able to adjust electron flow and maintain high photosynthesis under waterlogging conditions (Figure 4)." Line07 did not show significant differences with Line01 and Line21 after the waterlogging. If there are no significant differences, how can you consider that Line07 is able to adjust electron flow but Line01 is not? Moreover, why do you consider that values of 0.002 mmol m-2 s-1 CO2 uptakes are high?

Response:  Thanks for pointing this out. Yes, in fact, Line 27 is more tolerant than Line 21. The text had been revised to “Overall, the order of waterlogging tolerance of these four genotypes is: Line27 > Line07 > Line01 = Line21.” (Line 282).

You must critically discuss your results and compare them with other works. Figure 4 is poorly discussed in section 3.4. Moreover, you explain your results with data that you decide not to show (disease infection susceptibility and adventitious roots presence) If you consider that those factors are important to justify your results, you have to include them.

Response:  We rewrite the discussion section in this version. We had included adventitious roots in this version (section 2.6). Thanks for the good suggestion.   

You did not answer/justify the disagreements found in your results. In Figure 5C, gs = 0.04 mol m-2 s-1 and you considered that it is a high stomatal conductance, while in Figure 3B you stated for a similar value that stomata were closed. 

Response:  We confess that the lack of control in the water depth experiment caused a discrepancy in this study. Therefore, we decided to delete the water depth data in this version.     

L286 "Every increase of one meter in water depth reduces the gas saturation by 10%". You work in a range between 10 and 20 cm depth. You should also discuss papers that show similar conditions to those described in your experiment. Again, you justify your results according to an impaired root growth when you did not show any root development data. In my opinion, as you said that you are interested in studying Sacha inchi's root development, you should wait until you get your results to confirm or dispel your hypothesis.

Response:  We had added AR data in section 2.6. Adventitious root formation of sacha inchi after long-term waterlogging (Figure 5). We briefly discussed the importance of disease resistance under waterlogging in Line 307, “Therefore, in addition to abiotic stresses (high temperature and high light), the plant-microbe interactions should pay more attention.”   

Your answer to the question: if we compare the gs values reported in Figure 1 for the Line 07 at 32ºC, light intensity 1 mmol m-2 s-1, CO2 0.4 mmol mol-1, and RH 60 % during the period from 9 am to 3 pm, all the values reported are higher than 0.27 mol m-2 s-1. However, in Figures 2 and 3, the mean value provided is lower than 0.22 mol m-2 s-1. And in Figure 5c the gs mean value reported for Line 07, before the experiment, is 0.15 mol m-2 s-1. How can you explain that?

"The variations of gs values might due to different batches of experiment and environmental factors in the greenhouse. Normally, fluctuation of stomatal conductance is greater than CO2 uptake."

If you consider that the conditions in the greenhouse varied so much, you must show/discuss them. Which conditions did change and how did they change? Your gs values for the same seedling (Line07) and measuring conditions decreased by 40 %. 

Response:  We agree to the discrepancy in the experiment of water depth and decided to delete the water depth result in this paper. Thanks.  

Reviewer 2 Report

The MS has been improved by the authors. However, the authors are still unable to connect the dots in view of literature to draw the sound objectives. Thus the Introduction section is somewhat haphazard. From the revised version, it seems that the authors tends to assess photosynthetic changes in plants due to diurnal changes in light and temperature, at a range of temperature, at a range of light intensities and under waterlogged conditions. Assessment of photosynthetic capacity under waterlogged conditions is not linked with its other experiments. In addition, it is essential to assess/discuss which limitation prevails (Stomatal/metabolic) in Sacha inchi plants under different type of abiotic stresses (light, temperature and waterlogging) which may give some useful meaning/outcome from the MS.     

Author Response

Reviewer#2

The MS has been improved by the authors. However, the authors are still unable to connect the dots in view of literature to draw the sound objectives. Thus the Introduction section is somewhat haphazard. From the revised version, it seems that the authors tend to assess photosynthetic changes in plants due to diurnal changes in light and temperature, at a range of temperature, at a range of light intensities, and under waterlogged conditions. Assessment of photosynthetic capacity under waterlogged conditions is not linked with its other experiments. In addition, it is essential to assess/discuss which limitation prevails (Stomatal/metabolic) in Sacha inchi plants under different type of abiotic stresses (light, temperature and waterlogging) which may give some useful meaning/outcome from the MS.     

Response:  Thanks for your comments and good suggestions. We had made a major revision and rewritten the discussion section in this version.

The objective had been changed to “In this study, we characterized the photosynthetic behavior of sacha inchi under different conditions (diurnal, temperature, light), and analyzed the relationship between photosynthetic capacity and waterlogging tolerance. We also screened seedlings under long-term waterlogging conditions and investigated the morphological changes.” (Line 87-91).

  • We discussed more different types of abiotic stresses (light, temperature and waterlogging), to read “Figure 3c shows that there was no significant change in the transpiration rate at 32°C with exposure to different light intensities. The transpiration rate remained less than 4 mmol m−2s−1, even at high-intensity irradiation (3000 µmol m−2s−1). However, temperature significantly affected the transpiration rate. When the temperature was higher than 36°C, the transpiration rate was over 4 mmol m−2s−1 (Figure 2c). The high transpiration might cause severe water loss. The effect of temperature on transpiration of sacha inchi is more pronounced than light intensity. This may explain why, during summer, when there was waterlogging caused by heavy rain, accelerated wilting and death of sacha inchi was observed in the field. The dual effects of root damage and leaf water loss may lead to vulnerability. ” (Line 245-254)  

Reviewer 4 Report

The authors did not provide good responses. They essentially changed the wording of the manuscript but they could not address major concerns regarding the methods. I believe that this manuscript does not have the quality to be published so I do not endorse its publication. 

Author Response

The authors did not provide good responses. They essentially changed the wording of the manuscript but they could not address major concerns regarding the methods. I believe that this manuscript does not have the quality to be published so I do not endorse its publication. 

Response: Thanks for your time and comments. We had made major revisions in this new version and improved the readability of this article.

Round 3

Reviewer 1 Report

I appreciate the changes done to the manuscript by the authors, they have considerably improved the paper. However, I consider that the manuscript still has weak points that must be improved. Overall, the results and the discussion of the results must be improved according to the data shown.
Authors should consider comments below, among others, when they are improving the manuscript: 

L95. Please justify, why did you decide to maintain the temperature, RH and light intensity stable if you were interested in knowing how sacha inchi's gas exchange varies throughout the day. I cannot see the point of assessing the diurnal changes in photosynthesis when the environmental conditions are fixed.

L98. Why do you only show the results of Line07? Please, show the 3 Lines (Line 08-10 and 07).

L116. In this case, it makes sense to fix the light intensity in order to assess the effect of the temperature.

L173. The leaf wilting percentage of Lines 01 and 21 was more than 50% but there was less wilting in Lines 07 and 21 at 5 DAF. Please rewrite this sentence. It is not clear which was the response of the Line 21.

L178. You must state that the wilting percentage increased after the recovery and why.

L196. In the figures i and k, the results for the Line01 Wilt%_5DAF and Wilt%_Re7d are said to be statistically different. However, SEMs of both means overlap. Please, check if it is correct.

L226. Which Lines are each figure?

L237. In this paragraph you summarized your results but you did not discuss them. Please, discuss your results. 

L250. You must compare your results with other studies published with other plants similar to sacha inchi. Did sacha inchi show a similar or a different behavior under those abiotic stresses?

L266. Liu et al. [23] studied

L279. "After recovery, the leaf wilting percentage tends to decrease." The discussion makes sense but is contradictory with the results previously exhibited. In the figure 4k, the mean values of percentage of wilting leaves after recovery for 7d are higher for all the lines than after the waterlogging.

L346. Based on the diurnal change in photosynthesis data, we know sacha inchi tends to close stomata after 17:00. Your conditions cannot be extrapolated to the real diurnal changes.

Author Response

Review#1

Comments and Suggestions for Authors

I appreciate the changes done to the manuscript by the authors, they have considerably improved the paper. However, I consider that the manuscript still has weak points that must be improved. Overall, the results and the discussion of the results must be improved according to the data shown.
Authors should consider comments below, among others, when they are improving the manuscript: 

Response:  We sincerely appreciate your careful reading and comments for the improvement of this article. We have made the corrections accordingly to your guidance. Thanks again.

  1. Please justify, why did you decide to maintain the temperature, RH, and light intensity stable if you were interested in knowing how sacha inchi's gas exchange varies throughout the day. I cannot see the point of assessing the diurnal changes in photosynthesis when the environmental conditions are fixed.

Response:  In this study, we would like to know the potential of CO2 uptake and stomatal conductance of sacha inchi and fix the light intensity. We agree it is not “natural” sunlight and not suitable to use the term “Diurnal change”. We had revised to, “2.1. Variation of photosynthesis rate

To understand the potential of CO2 uptake and stomatal conductance of sacha inchi, we used a Li-Cor 6800 photosynthesis system to detect CO2 uptake and stomatal conductance of sacha inchi two lines (Line 08-10 and Line 07) under constant 1000 µmol m−2s−1 irradiation to avoid sunlight fluctuation in the greenhouse. The results indicated that Line 07 has higher CO2 uptake and stomatal conductance (gs) than Line 08-10, but both lines have a similar changing trend (Figure 1). CO2 uptake of Line 07 was moderately high at 06:00 (8 μmol m-2s-1), high at 08:00 to 15:00 (12.5 μmol m-2s-1), and then declined to less than 3 μmol m-2s-1 at 20:00 (Figure 1a). The stomatal conductance of sacha inchi was less than 0.1 mol m-2s-1 in the early morning and after 18:00. During the daytime from 09:00 to 14:30, it had a high stomatal conductance value (>0.3 mol m-2s-1) (Figure 1b). These results indicate that the stomatal opening of sacha inchi is affected by circadian rhythm, which therefore affects the CO2 uptake. Moreover, there is genetic variation in photosynthesis capacity.” (L.95-117)

  1. Why do you only show the results of Line07? Please, show the 3 Lines (Line 08-10 and 07).

Response: In the beginning, we just showed Line 07 because it had been recorded with a longer time course and had higher CO2 uptake than Line 08-10. We had included data of both lines (07 and 08-10) in this new version. Revised “Figure 1. The variation of CO2 uptake and stomatal conductance of sacha inchi seedlings: (a) CO2 uptake and (b) stomatal conductance of sacha inchi Line 07 and Line 08-10 seed-lings. Li-Cor chamber conditions were: temperature, 32°C; light intensity, 1000 µmol m−2s−1; CO2, 400 µmol mol−1; and RH, 60%. An auto logging program was used to record data every 60 seconds. Line 07 was recorded from 06:00 to 19:50 but Line 08-10 was recorded from 08:15 to 19:50.” (L.112-117)         

L116. In this case, it makes sense to fix the light intensity in order to assess the effect of the temperature.

Response:  OK

  1. The leaf wilting percentage of Lines 01 and 21 was more than 50% but there was less wilting in Lines 07 and 21 at 5 DAF. Please rewrite this sentence. It is not clear which was the response of Line 21.

Response: Thanks for pointing out the redundancy of Line 21. It is rewritten to “ The leaf wilting percentage of Lines 01 and 21 was more than 50% but there was less wilting in Lines 07 and 27 at 5 DAF. “ (L.176-177).

  1. You must state that the wilting percentage increased after the recovery and why.

Response: OK, we had revised to “After recovery in the greenhouses for 7 days, only Line 27 had less than 50% leaf wilting. Statistical analysis showed that the wilting percentage of the four Lines was not significantly different between 5 DAF and after recovery for 7 days, however, the mean wilting percentage of the Lines 01, 07, and 21 were increased after recovery (Figure 4k).” (L.178-181).

In the discussion section, we explained more, “Except for Line 27, Lines 01, 07, and 21 showed severe wilting, which may be due to ROS injury, and their root function cannot be restored after recovery. Flooding recovery causes a sudden oxygen burst that accelerates oxidative stress, increases plant damage, and leads to rotten roots and wilting leaves had been reported [19,26].” (L.290-294)

  1. In the figures i and k, the results for the Line01 Wilt%_5DAF and Wilt%_Re7d are said to be statistically different. However, SEMs of both means overlap. Please, check if it is correct.

Response:  We had checked and revised Fig. 4 i and k. Thanks

  1. Which Lines are each figure?

Response:  We explained in section 4.1 “To screen the long-term waterlogging tolerance line, about 200 bulk seeds collected from the farmer’s field were sown, and the seedlings were waterlogged for two weeks. (L. 347-348)

  1. In this paragraph you summarized your results but you did not discuss them. Please, discuss your results. 

Response:  We had included more discussion section 3.1, “Presumably, the increase sacha inchi transpiration rate seen with high-temperature is mainly due to the effect of temperature-dependent cuticle transpiration as mentioned previously [20,21]. It was reported that when temperatures are higher than 35°C, there is significantly increased cuticular water permeability. Therefore, to improve sacha inchi nursery quality, it is suggested that greenhouse environments are maintained at the above suitable conditions to increase seedlings’ photosynthesis rate and enhance growth. High transpiration might cause severe water loss. We found that the effect of temperature on transpiration of sacha inchi is more pronounced than light intensity. This may explain why, during summer, when there is waterlogging caused by heavy rain, there is tremendous wilting and the death of sacha inchi in the field. The dual effects of root damage and leaf loss of water may lead to vulnerability. ” (L.251-261)

  1. You must compare your results with other studies published with other plants similar to sacha inchi. Did sacha inchi show a similar or a different behavior under those abiotic stresses?

Response: OK, the revision is included in Response#7. Thanks

  1. Liu et al. [23] studied

Response: OK, it has been revised to Liu et al. [23] studies…. Thanks.   

  1. "After recovery, the leaf wilting percentage tends to decrease." The discussion makes sense but is contradictory with the results previously exhibited. In the figure 4k, the mean values of percentage of wilting leaves after recovery for 7d are higher for all the lines than after the waterlogging.

Response: We specified ‘Line 27’ in this new version. “After recovery, the leaf wilting percentage of Line 27 tends to decrease, and the seedlings can grow and produce new leaves.” (L.285-287). We also added “Except for Line 27, Lines 01, 07, and 21 showed severe wilting, which may be due to ROS injury, and their root function cannot be restored after recovery. Flooding recovery causes a sudden oxygen burst that accelerates oxidative stress, increases plant damage, and leads to rotten roots and wilting leaves had been reported [19,26].” (L.290-294). Thanks.

  1. Based on the diurnal change in photosynthesis data, we know sacha inchi tends to close stomata after 17:00. Your conditions cannot be extrapolated to the real diurnal changes.

Response:  It is retyped to “Based on the variation of photosynthesis, we know sacha inchi tends to close stomata after 17:00. “ (L.358). Thanks